# Sustainable Intensification of Sorghum and Pearl Millet Production by Seed Priming, Seed Treatment and Fertilizer Microdosing under Different Rainfall Regimes in Mali

**Adama Coulibaly [1], Kamkam Woumou [1] and Jens B. Aune [2],***

[1] Institut d'Economie Rurale, IER, LABOSEP de Sotuba, BP. 262, Bamako, Mali;
adamacz097@gmail.com (A.C.); kamkamwoumou2002@yahoo.fr (K.W.)

[2] Department of International Environment and Development Studies (Noragric),
Faculty of Landscape and Society, Norwegian University of Life Sciences, 1432 Aas, Norway

\* Correspondence: jens.aune@nmbu.no; Tel.: +47-67231318

**Abstract:** Sorghum and pearl millet are the most important stable crops in the drylands of West Africa. This four-year study based on two experiments assessed different low-cost methods for intensification of sorghum and millet production under contrasting rainfall conditions in Mali. Experiment 1 assessed the microdosing rates 0, 0.2, 0.4, 0.6 and 0.8 g NPK hill$^{-1}$ combined with seed priming across four locations in Mali while Experiment 2 assessed the cumulative effects of the seed priming, seed treatment with pesticide, microdosing and urea top dressing. In central Mali, there was a clear effect of seed priming while there was no such effect in southern Mali with better rainfall conditions. In central Mali, the microdosing rate of fertilizer of 0.4 g NPK hill$^{-1}$ (4 kg NPK ha$^{-1}$) performed best, while in southern Mali the microdosing rate of 0.8 g NPK hill$^{-1}$ (20 kg NPK ha$^{-1}$) gave the best result. Experiment 2 showed that there was a clear effect of top dressing of 1 g urea hill$^{-1}$ (25 kg urea ha$^{-1}$) in southern Mali while no such effect was apparent in central Mali. In general, there was a better response to microdosing in sorghum than in pearl millet. A decision tree for crop and fertilizer management in Mali was developed, taking into consideration rainfall, crop choice, use of seed priming and microdosing rates.

**Keywords:** seed priming; microdosing; seed treatment; economic return; fertilizer use efficiency; value–cost-ratio; decision tree for crop and fertilizer management

## 1. Introduction

The Sahelian countries, including Mali, face a tremendous challenge to produce sufficient food for the rapidly growing population of these countries. For Mali, it has been calculated that cereal production will need to increase by 365% between 2010 and 2050 in order to keep pace with population growth [1]. To achieve such an increase in food production will be very challenging owing to climate change, highly variable rainfall, low soil fertility, limitation in infrastructure and political instability [2–5]. Studies on the effect of climate change on agricultural productivity in the tropics show that without interventions to address climate change, food production is likely to decline [6]. Poverty, risk of crop failure, variable prices of harvest and input and uncertain access to credit are difficulties farmers face, compelling them to make suboptimal decisions that result in risk aversion behaviour, which keeps farmers in a poverty trap.

Sorghum (*Sorghum bicolor* (L.) Moench) and pearl millet (*Pennisetum glaucum* (L.) R. Br.) are the major stable crops that the farmers grow in the Sahel. However, their production is limited by factors

such as erratic rainfall, low soil fertility and pests and diseases [3,7]. The crops are often sown in hills (clusters of seeds). The distance between the hills can be beyond 1 m corresponding to a planting density below 10,000 hills ha$^{-1}$. Farmers have traditionally maintained soil fertility by manuring and fallowing. Use of mineral fertilizer has been low in sorghum and pearl millet.

Farmers will look for production methods with a low cost, but a high return. Such technologies can be considered "low-hanging fruit" for agricultural development. Microdosing, developed in the 1990s by the International Crop Research Institute for the Semi-Arid Tropics (ICRISAT), can be characterized as one such low cost method of agricultural intensification. The ICRISAT recommendation was to place mineral fertilizer at a rate of 2 g diammonium phosphate (DAP) or 6 g NPK fertilizer hill$^{-1}$ corresponding to fertilizer rates of 20 kg DAP ha$^{-1}$ or 60 kg NPK ha$^{-1}$ when there are 10,000 planting hills ha$^{-1}$ as commonly practised in Niger [8]. However, in pearl millet and sorghum producing areas in southern Mali farmers commonly use 25,000 planting hill ha$^{-1}$, corresponding to fertilizer application rates of 50 kg DAP ha$^{-1}$ and 150 kg NPK ha$^{-1}$ if the ICRISAT rates are used. These rates represent a cash outlay beyond the reach of most farmers.

Research in the first decade of 2000 identified different ways to practise microdosing [9]. In Niger, it was shown that it is possible to delay the fertilizer application up to 20 days after sowing without causing a yield penalty [9]. In Sudan and Mali, rates as low as 0.3 g fertilizer hill$^{-1}$ can increase yield by 20–50% [10,11]. It was concluded based on a study in Niger that the microdosing rates recommended by ICRISAT are risky, as many farmers will get a negative economic return [12]. Fertilizer rates in the order of 2 g hill$^{-1}$ and beyond may burn the seeds particularly if the seeds and the fertilizer is mixed. Seeds and fertilizer must therefore be applied in two operations if such rates of fertilizer are applied thereby increasing the labour demand for the farmers and delaying the sowing process.

When rates as low as 0.3 g fertilizer hill$^{-1}$ are applied, it is possible to mix seed and fertilizer in a 1:1 ratio without causing burning of the seeds [10]. This allows for easy manual application or simultaneous application of seeds and fertilizer using a planter [13].

Seed priming, which consists of soaking seed of millet and sorghum in water for eight hours prior to sowing, is another low-input agricultural technology for improving crop establishment and yield under dryland conditions [11,14]. Combining seed priming with increasing rates of microdosing has previously not been studied in the drylands of West Africa. This study therefore set out to identify appropriate microdosing rates in sorghum and pearl millet in combination with seed priming. Seed priming is particularly important in order to achieve a rapid and uniform crop establishment [14]. The low-cost technologies will not realize the full potential yield of these crops in the drylands of West Africa. The objective was rather to identify a low-cost entry point for agriculture intensification, and to identify under which conditions these methods will work the best.

The effects of applying rates beyond 2 g fertilizer hill$^{-1}$ are well documented in previous research [7]. However, to our knowledge, no systematic tests have been conducted to identify appropriate NPK microdosing rates in the range 0.2–0.8 g NPK hill$^{-1}$ for dryland cereals in West Africa. In addition, this research assessed the effect of combining microdosing with other yield enhancing technologies such as seed priming, seed treatment with fungicide/insecticide and top dressing of urea. Farmers are very cash constraint in the drylands of West Africa and it was therefore of great research interest to also assess rates in the lower end of the response curve, even though these rates will not realize the full yield potential of the crops.

## 2. Materials and Methods

Experiment 1 tested seed priming (with and without) with increasing levels of microdosing while Experiment 2 assessed the cumulative effect of introducing seed priming, seed treatment, microdosing (0.3 g NPK hill$^{-1}$) and top dressing of urea. These experiments were conducted in the period 2011–2014.

## 2.1. Site Characteristics

The experiments were undertaken in four locations as shown in Table 1. The Sotuba site (Bamako) was located in the Sahelo-Sudanian climate zone, while the sites in Madina Kagoro, Koporo-Pen and Koro were located in the Sahelian zone [15].

**Table 1.** Characteristics of the four sites hosting the experiments.

| | Location | Zone and Rainfall | Crop Tested | Experiment |
|---|---|---|---|---|
| Sotuba Bamako | 12°65′60″ N 7°92′54″ W | Sahelo–Sudanian, 850–900 mm | Sorghum, pearl millet | 1 and 2 |
| Madina Kagoro Koulikoro region | 12°35′50″ N 7°51′31″ W | Sahelian, 350–400 mm | Sorghum | 1 |
| Koporo-Pen Mopti region | 14°8′7.20″ N 3°11′11″ W | Sahelian, 450 mm | Pearl millet | 1 and 2 |
| Koro Mopti region | 14°7′51″ N 3°21′14″ W | Sahelian, 450 mm | Pearl millet | 1 |

The soil at Koporo was a sandy soil, while in Sotuba the soil could be characterized as a loamy sand. Both soils had very low levels of soil organic carbon and available phosphorous (Table 2).

**Table 2.** Soil properties at the experimental sites in Sotuba and Koporo-Pen at soil depth of 0–20 cm.

| Variable | Sotuba (*n* = 20) | Koporo-Pen (*n* = 20) |
|---|---|---|
| pH (water) | 5.7 | 5.4 |
| Soil organic carbon % | 0.14 | 0.27 |
| Available phosphorous mg/kg | 1.85 | 2.9 |
| CEC meq/100 g | 2.89 | 2.78 |
| Sand % > 0.05 mm | 79.6 | 92.0 |
| Silt % 0.05–0.002 mm | 17.15 | 3.8 |
| Clay % < 0.002 mm | 3.15 | 4.2 |

CEC is Cation Exchange Capacity.

During the four years of the study, rainfall in Sotuba varied from 866 to 1016 mm with an average of 919 mm whereas in Koporo-Pen rainfall varied from 389 to 599 mm with an average of 478 mm. In Madina Kagora, the average rainfall for the two years was 509 mm with a variation from 345 to 673 mm.

## 2.2. Planting Material

The sorghum varieties Tiandougou (growth cycle 120 days) and CSM63 E (growth cycle 100 days) were used in Sotuba and Madina Kagoro respectively. The pearl millet variety used was Sanio (growth cycle 110 days) in Sotuba, and Toroniou (growth cycle 110 days) in the Mopti region.

When seed priming was used as a treatment, the seeds were cleaned by winnowing and sieving, followed by soaking the seeds in water for eight hours. Seeds were dried for two hours in shadow prior to sowing in order to reduce the stickiness of the seeds, thereby facilitating sowing.

The planting geometry in sorghum was $80 \times 50$ cm for both sites, giving a density of 25,000 hills ha$^{-1}$. For pearl millet, the distance between hills was the same as for sorghum in Sotuba, while the planting distance was $100 \times 100$ cm in the Mopti region, which corresponded to a density of 10,000 hills ha$^{-1}$.

The plants were thinned to 2–3 plants hills$^{-1}$. The plot size in both experiments was 24 m$^2$ and the harvested area was 4.9 m$^2$.

## 2.3. Experiment 1

The objective of this experiment was to test the effect of combining different rates of microdosing of NPK fertilizer with and without seed priming. The fertilizer used was the compound and granulated NPK 15–15–15 (produced by Société Tropicale d'Engrais et de Produits Chimiques), applied in the

same hill as the seeds by using a measuring cup delivering the amount of fertilizer as described in the treatment. The plots were fixed for four years and there were five replications. The experiment was a split plot design with seed priming on the large plots (with and without) and the microdosing rates on the smaller plots. The microdosing rates were 0, 0.2, 0.4, 0.6 and 0.8 g hill$^{-1}$ corresponding to 0, 5, 10, 15 and 20 kg ha$^{-1}$ when using 25,000 hills$^{-1}$. The above rates per hill corresponded to 0, 2, 4, 6 and 8 kg NPK ha$^{-1}$ when planting density was 10,000 hills ha$^{-1}$.

### 2.4. Experiment 2

The objective of this experiment was to assess a pathway for intensification by studying the cumulative effects of different inputs (seed priming, seed treatment with pesticide, microdosing and top dressing) arranged according to the increasing cost of the input.

The results of this experiment could assist farmers to determine an intensification level that corresponds to their resource endowment. This was an on-station experiment conducted at Sotuba research station in Bamako and at the sub-research station at Koporo-Pen in the Mopti region.

The following treatments were included:

1. Control.
2. Seed priming (SP).
3. SP + seed treatment (ST).
4. SP + ST + microdosing 5 kg NPK ha$^{-1}$ (M).
5. SP + ST + M + microdosing 5 kg NPK ha$^{-1}$ at tillering (MT).
6. SP + ST + M + microdosing 25 kg urea ha$^{-1}$ at tillering.

The seed treatment consisted of treating the seeds with Caiman rouge (2.5 g per kg seed) which is a combined insecticide/fungicide consisting of Permethrin 25 g kg$^{-1}$ and Thiram 250 g kg$^{-1}$. The cost of seed treatment was 250 FCFA ha$^{-1}$ (FCFA = Franc Communauté Financière d'Afrique).

### 2.5. Measurements and Data Analysis (Common to the Two Experimental Series)

Observations were taken on number of established plants, number of harvested heads and grain, and stover yield. A statistical analysis was undertaken using the general linear model in the SAS statistical software and the Duncan test was used to differentiate between the treatments.

Fertilizer use efficiency (FUE) was calculated by dividing the increased grain yield due to fertilizer by the amount of fertilizer applied [12]. The threshold value was 10 kg of grain (millet or sorghum) for each kg of fertilizer [16]. The FUE was calculated for the combined effect of seed priming and microdosing and for microdosing alone.

### 2.6. Soil Analysis

Soil parameters measured were pH, carbon, available phosphorus, cationic exchange capacity and texture (Table 2). The pH was determined in water solution with 1/2.5 ratio with the potentiometric method by using a pH metre [17]. Carbon was determined using a modified dichromate method at 120 °C [18]. The Bray I method [19] was used to determine available phosphorus. Cationic exchange capacity was determined using ammonium acetate at pH 7 [20]. The pipet method was used to determine soil texture for silt, sand and clay [21].

### 2.7. Economic Analysis

A partial gross margin for each treatment was calculated as the difference between the revenue from the grains and the cost. The cost items were inputs and the labour related to seed priming and microdosing. The other labour costs were not included. Local prices for the grains and input were used.

## 3. Results

The results are presented separately for each experiment.

### 3.1. Effect of Microdosing Rates Combined with Seed Priming (Experiment 1)

At Sotuba, there was no effect of seed priming in sorghum on the diameter of straw and yield (Table 3). However, priming slightly increased the number of germinated planting hills. There was a strong effect of increasing microdosing rates yield. Microdosing increased the number of hill ha$^{-1}$ and straw diameter in sorghum, and the application of 20 kg NPK ha$^{-1}$ compared to the control increased stover and grain yield by 6734 kg ha$^{-1}$ (78.3%) and 638 kg ha$^{-1}$ (52.3%) respectively. A rate as low as 10 kg fertilizer ha$^{-1}$ gave 28% higher grain yield than the control. As seen in Table 3, the fertilizer use efficiency was above 30 for all the microdosing treatments, but there was no decreasing trend with increasing the rate of microdosing. There was no interaction between priming and microdosing on the measured parameters.

**Table 3.** Effect on sorghum of seed priming and increasing microdosing rates on number of hills ha$^{-1}$, diameter of straw, stover yield, grain yield, gross margin, value–cost-ratio (VCR) and fertilizer use efficiency (FUE) for Sotuba, Bamako (average for 2011–2014).

| Treatments | Germ. Hills ha$^{-1}$ | Diam. Straw 30 DAS mm | Diam. Straw 50% Flowering mm | Stover Yield kg ha$^{-1}$ | Grain Yield kg ha$^{-1}$ | Gross Margin FCFA ha$^{-1}$ | VCR | FUE |
|---|---|---|---|---|---|---|---|---|
| Priming | | | | | | | | |
| Without Priming | 20,590 b | 8.3 | 22.4 | 11,810 | 1520 | 108,800 | 9.64 | 21.8 |
| Primed Seeds | 21,340 a | 9.1 | 23 | 11,800 | 1590 | 115,900 | 15.56 | 35.6 |
| SE ± | 244 | 0.28 | 0.27 | 377 | 56.3 | 5624.5 | 3.05 | 6.56 |
| Probability | 0.036 | 0.063 | 0.21 | 0.98 | 0.38 | 0.38 | 0.18 | 0.14 |
| Microdoses | | | | | | | | |
| 0 kg ha$^{-1}$ | 19,950 a | 7 a | 21 a | 8596 a | 1218 a | 78,700 a | - | - |
| 5 kg ha$^{-1}$ | 21,260 b | 8 ab | 22 ab | 10,850 b | 1456 ab | 102,500 ab | 20.1 a | 47.1 a |
| 10 kg ha$^{-1}$ | 21,200 b | 9.1 bc | 23 b | 11,500 bc | 1560 b | 112,900 b | 15.0 a | 33.9 a |
| 15 kg ha$^{-1}$ | 21,070 b | 9.6 c | 23 b | 12,760 c | 1684 bc | 125,300 bc | 13.9 ab | 30.8 a |
| 20 kg ha$^{-1}$ | 21,340 b | 10 c | 25 c | 15,330 d | 1856 c | 142,400 c | 14.0 ab | 31.6 a |
| SE ± | 385 | 0.44 | 0.42 | 596 | 8893 | 8893.1 | 4.83 | 10.4 |
| Probability | 0.004 | 0.0001 | 0.0001 | 0.0001 | 0.0001 | 0.0001 | 0.06 | 0.039 |
| Priming × Microdose | 0.64 | 0.65 | 0.91 | 0.59 | 0.84 | 0.85 | 0.74 | 0.65 |

Numbers followed by different letters are statistically different treatments according to Duncan multiple rank test. DAS = days after sowing, FCFA = West African CFA franc.

Seed priming did not increase gross margin compared to the control while the application of 20 kg NPK ha$^{-1}$ increased the gross margin by 80.9% compared to the control. The value–cost-ratio (VCR) was between 14 and 20 for all the treatments and the FUE varied between 30 and 47 for all the treatments. The VCR and FUE decreased with increasing fertilizer rates, but this effect was not significant.

There was no effect of seed priming on crop development parameters, stover and grain yield or on gross margin in pearl millet in Sotuba (Table 4). Increasing the rates of microdosing increased the yield, but the yield increase was not as strong as in sorghum. Application of 20 NPK ha$^{-1}$ compared to the control increased stover and grain yield by 1430 (16.3%) and 187 kg ha$^{-1}$ (17.8%) respectively. Only the highest microdosing rate gave a grain yield that was significantly different from the control. The fertilizer use efficiency was 10.25 for the 5 kg NPK ha$^{-1}$ treatment. The interaction between seed priming and microdosing was not significant.

Seed priming did not increase the yield gross margin for pearl millet in Sotuba, while the application of 20 kg NPK hill$^{-1}$ increased the gross margin by 30.1%. For NPK rates, the value-cost-ratio varied between 0.13 and 4.75 while the FUE varied between 0.25 and 10.25.

**Table 4.** Effect on pearl millet of seed priming and increasing microdosing rates on number of hills ha$^{-1}$, diameter of straw, stover yield, grain yield, gross margin, value–cost-ratio (VCR) and fertilizer use efficiency (FUE) for Sotuba, Bamako (Average for 2011, 2012, 2013 and 2014).

| Treatment | Germ. Hills ha$^{-1}$ | Diam. Straw 30 DAS mm | Diam. Straw 50% Flowering mm | Stover Yield kg ha$^{-1}$ | Grain Yield kg ha$^{-1}$ | Gross Margin FCFA ha$^{-1}$ | VCR | FUE |
|---|---|---|---|---|---|---|---|---|
| Priming | | | | | | | | |
| Without Priming | 21,380 | 9.5 | 10.9 | 9429 | 1082 | 65,020 | 3.3 | 3.60 |
| Primed Seeds | 20,830 | 9.3 | 10.6 | 10,200 | 1137 | 70,550 | 7 | 5.55 |
| SE ± | 212.5 | 0.19 | 0.15 | 354.5 | 35.35 | 3533.6 | 2.75 | 3.36 |
| Probability | 0.22 | 0.36 | 0.24 | 0.14 | 0.28 | 0.65 | 0.35 | 0.92 |
| Microdoses | | | | | | | | |
| 0 kg ha$^{-1}$ | 20,340 a | 8.25 a | 10.4 a | 8750 a | 1051 a | 61,940 a | - | - |
| 5 kg ha$^{-1}$ | 21,350 a | 9.63 b | 10.5 a | 10,170 a | 1103 a | 67,180 ab | 4.75 a | 10.25 a |
| 10 kg ha$^{-1}$ | 21,410 a | 9.5 b | 11 a | 9990 a | 1053 ab | 62,140 a | 0.13 a | 0.25 a |
| 15 kg ha$^{-1}$ | 21,530 a | 10 b | 11 a | 9983 a | 1102 ab | 67,050 ab | 4.25 a | 3.38 a |
| 20 kg ha$^{-1}$ | 20,680 a | 10 b | 11 a | 10,180 a | 1238 b | 80,600 b | 4.0 a | 9.00 a |
| SE ± | 494 | 0.3 | 0.24 | 560 | 56 | 5587 | 2.33 | 5.32 |
| Probability | 0.5 | 0.003 | 0.24 | 0.35 | 0.15 | 0.14 | 0.48 | 0.52 |
| Priming × Microdose | 0.96 | 0.35 | 0.12 | 0.90 | 0.65 | 0.65 | 0.88 | 0.92 |

Numbers followed by different letters are statistically different treatments according to Duncan multiple rank test. DAS = days after sowing, FCFA = West African CFA franc.

In Madina Kagora, 200 km north of Sotuba/Bamako, seed priming increased stover and grain yield of sorghum by 837 kg ha$^{-1}$ (28.6%) and 184 kg ha$^{-1}$ (19.7%) respectively (Table 5). At this site, seed priming also significantly increased the diameter of straw, but did not significantly increase the number of germinated hills per hectare. Microdosing of 20 kg NPK ha$^{-1}$ compared to the control increased stover and grain yield by 2354 (123.7%) and 690 kg ha$^{-1}$ (103.4%) respectively. A rate as low as 5 kg ha$^{-1}$ significantly increased grain yield over the control by 214 kg ha$^{-1}$ (32.1%). Microdosing did not increase the number of planting stations at the crop establishment phase or at harvest, but the diameter of straw increased with increasing microdosing rates. As a result of the increased stover and grain yield, there was a strong effect on the gross margin. Seed priming increased the gross margin by 36.6% while the application of 20 kg NPK ha$^{-1}$ increased the gross margin by 3.9 times. VCR varied between 15 and 19 with the highest VCR obtained for application of 5 kg NPK ha$^{-1}$. The FUE varied between 34 and 42 for the treatments.

**Table 5.** Effect of seed priming and increasing microdosing rates on number of plants hill$^{-1}$, plant vigour, stover yield, grain yield, gross margin, value–cost-ratio (VCR) and fertilizer use efficiency (FUE) of sorghum at Madina Kagoro (Average for the years 2011–2012).

| Treatment | Germ. Hills ha$^{-1}$ | Diam. Straw 30 DAS mm | Diam. Straw 50% Flowering mm | Harvested Planting Stations ha$^{-1}$ | Stover kg ha$^{-1}$ | Grains kg ha$^{-1}$ | Gross Margin FCFA ha$^{-1}$ | VCR | FUE |
|---|---|---|---|---|---|---|---|---|---|
| Priming | | | | | | | | | |
| Without Priming | 21,210 | 9.4 | 14.2 | 19,420 | 2930 | 933.6 | 50,200 | 12.96 | 29.0 |
| Primed Seeds | 21,320 | 8.3 | 14 | 20,920 | 3769 | 1117 | 68,590 | 13.2 | 29.5 |
| SE ± | 310.4 | 0.29 | 0.19 | 817.57 | 203.32 | 41.64 | 4163.68 | 1.91 | 4.24 |
| Probability | 0.81 | 0.086 | 0.024 | 0.20 | 0.006 | 0.003 | 0.003 | 0.93 | 0.94 |
| Microdoses | | | | | | | | | |
| 0 kg ha$^{-1}$ | 21,830 a | 7.1 a | 13.3 a | 17,500 a | 1908 a | 667 a | 23,570 a | - | - |
| 5 kg ha$^{-1}$ | 21,560 a | 8.4 ab | 13.4 a | 20,620 a | 3050 b | 881 b | 44,950 b | 19.1 b | 42.4 a |
| 10 kg ha$^{-1}$ | 21,190 a | 9.2 bc | 13.8 ab | 21,040 a | 3472 bc | 1013 b | 58,170 b | 15.2 a | 34.2 a |
| 15 kg ha$^{-1}$ | 20,940 a | 9.6 bc | 14.1 ab | 21,040 a | 4056 c | 1210 c | 77,800 c | 15.9 a | 35.6 a |
| 20 kg ha$^{-1}$ | 20,790 a | 10.3 c | 14.6 c | 20,620 a | 4262 c | 1357 c | 92,490 c | 15.2 a | 34.1 a |
| SE ± | 490.82 | 0.46 | 0.31 | 1292.7 | 321.48 | 65.84 | 6583.36 | 3.02 | 6.7 |
| Probability | 0.55 | 0.0001 | 0.028 | 0.27 | 0.0001 | 0.0001 | 0.0001 | 0.001 | 0.001 |
| Priming × Microdose | 0.40 | 0.64 | 0.13 | 0.18 | 0.79 | 0.66 | 0.66 | 0.08 | 0.81 |

Numbers followed by different letters are statistically different according to Duncan multiple rank test. DAS = days after sowing, FCFA = West African CFA franc.

In Koporo-Pen in the Mopti region, there was also a significant effect of seed priming and microdosing on pearl millet (Table 6). Seed priming increased stover and grain yield compared to the control by 560 kg ha$^{-1}$ (17.6%) and 210 kg ha$^{-1}$ (26.1%) respectively. At this site there was no response to microdosing beyond 6 kg fertilizer ha$^{-1}$. The application of 6 kg NPK ha$^{-1}$ increased stover and grain yield compared to the control by 1686 (69.5%) and 375 kg ha$^{-1}$ (54.9%) respectively.

**Table 6.** Effect of seed priming and increasing microdosing rates on number of plants per pocket, plant vigour, stover yield, grain yield, gross margin, value–cost-ratio (VCR) and fertilizer use efficiency (FUE) of pearl millet at the research station Koporo-Pen, 2011–2014.

| Treatment | Harvested hills ha$^{-1}$ | Stover kg ha$^{-1}$ | Grain ha$^{-1}$ | Gross Margin FCFA ha$^{-1}$ | VCR | FUE |
|---|---|---|---|---|---|---|
| Priming | | | | | | |
| Without Priming | 9165 | 3175 | 805.20 | 58,370 | 16.85 | 37.5 |
| Primed Seeds | 9388 | 3735 | 1015 | 37,360 | 24.35 | 54.0 |
| SE ± | 213.65 | 158.23 | 43.36 | 4333.74 | 4.69 | 10.36 |
| Probability | 0.47 | 0.018 | 0.002 | 0.002 | 0.27 | 0.27 |
| Microdoses | | | | | | |
| 0 kg ha$^{-1}$ | 8719 a | 2423 a | 683 a | 25,180 a | - | - |
| 2 kg ha$^{-1}$ | 8772 ab | 3008 a | 787 ab | 35,590 ab | 23.5 a | 52 a |
| 4 kg ha$^{-1}$ | 9241 abc | 3789 b | 954 bc | 52,270 ab | 30.25 a | 67 a |
| 6 kg ha$^{-1}$ | 9862 c | 4109 b | 1058 c | 62,640 b | 27.88 a | 62 a |
| 8 kg ha$^{-1}$ | 9788 bc | 3947 b | 1068 c | 63,640 b | 21.38 ab | 48 a |
| SE ± | 337.81 | 250.18 | 68.56 | 6852.24 | 7.41 | 16.37 |
| Probability | 0.058 | 0.0001 | 0.001 | 0.001 | 0.05 | 0.052 |
| Priming × Microdose | 0.34 | 0.98 | 0.98 | 0.98 | 0.83 | 0.84 |

Numbers followed by different letters are statistically different according to Duncan multiple rank test. FCFA = West African CFA franc.

Seed priming increased gross margin by 56% while the microdosing rates from 4 kg NPK ha$^{-1}$ and beyond more than doubled the gross margin. The VCR was between 21 and 30 while the FUE varied between 48 and 67.

As in the other sites in central Mali, there was a strong effect of seed priming in Koro (Table 7). Here, seed priming compared to the control increased stover and grain yield by 413 kg ha$^{-1}$ (24.1%) and 110 kg ha$^{-1}$ (18.0%) respectively. Application of 6 kg fertilizer ha$^{-1}$ increased the stover and grain yield by 254 kg ha$^{-1}$ (14.3%) and 186 kg ha$^{-1}$ (34.4%) respectively. There was a significant effect on grain yield of microdosing from 4 kg fertilizer ha$^{-1}$ and beyond. Seed priming increased the gross margin by 70% and the application rate of 6 kg NPK ha$^{-1}$ increased the gross margin by 170%. There was a significant effect on the gross margin from 4 kg NPK ha$^{-1}$ and beyond. The VCR varied between 9.5 and 12.8 while the FUE varied between 21.7 and 31.8 with the lowest FUE obtained for 10 kg NPK ha$^{-1}$.

**Table 7.** Effect of seed priming and microdosing of mineral fertilizer on stover yield, grain yield, gross margin, value–cost-ratio (VCR) and fertilizer use efficiency (FUE) for pearl millet in Koro (Average for 2011–2014).

| Treatment | Stover kg ha$^{-1}$ | Grain kg ha$^{-1}$ | Gross Margin FCFA ha$^{-1}$ | VCR | FUE |
|---|---|---|---|---|---|
| Priming | | | | | |
| Without Priming | 1713 | 595.9 | 16,390 | 6.15 | 13.95 |
| Primed Seeds | 2126 | 706 | 27,360 | 13.95 | 31.35 |
| SE ± | 52.39 | 20.57 | 2057.62 | 2.24 | 5 |
| Probability | 0.0001 | 0.001 | 0.001 | 0.02 | 0.021 |
| Microdoses | | | | | |
| 0 kg ha$^{-1}$ | 1775 a | 541 a | 10,920 a | - | - |
| 2 kg ha$^{-1}$ | 1884 a | 599 ab | 16,720 ab | 12.88 b | 28.88 b |
| 4 kg ha$^{-1}$ | 1894 a | 669 bc | 23,650 bc | 14.25 b | 31.75 b |
| 6 kg ha$^{-1}$ | 2029 a | 727 c | 29,500 c | 13.63 b | 30.88 b |
| 8 kg ha$^{-1}$ | 2014 a | 718 c | 28,580 c | 9.5 ab | 21.75 a |
| SE ± | 82.84 | 32.53 | 3253.38 | 3.54 | 7.96 |
| Probability | 0.20 | 0.001 | 0.001 | 0.04 | 0.045 |
| Priming × Microdose | 0.97 | 0.85 | 0.85 | 0.38 | 0.41 |

Numbers followed by different letters are statistically different according to Duncan multiple rank test. DAS = days after sowing, FCFA = West African CFA franc.

## 3.2. Effects of Increasing Level of Intensification (Experiment 2)

In Experiment 2, the cumulative effect of introducing inputs according to increasing cost was studied. The inputs included in this test with step-wise intensification were seed priming, seed treatment with combined fungicide/insecticide, microdosing and top dressing of fertilizer. The magnitude of the response and the effect of the treatment varied between the sites.

In Sotuba/Bamako, there was no effect on sorghum of the treatments on germinated hills and diameter of straw at the different development stages (Table 8). There was a tendency that increasing the level of intensification increased the stover and grain yield; however, it was only the treatment with the highest level of intensification that produced a grain yield which significantly differed from the other treatments. For this treatment which included urea top dressing, the yield increase compared to the control for stover and grain yield was 6900 (67.6%) and 893 kg ha$^{-1}$ (76.1%) respectively. The gross margin was increased by 91.8%. The VCR for the treatments varied between 12.8 and 15.2 and the fertilizer use efficiency was between 29 and 41 kg grain kg$^{-1}$ fertilizer.

**Table 8.** Effect of increasing level of intensification on crop development, yield parameters, gross margin, value–cost-ratio (VCR) and fertilizer use efficiency (FUE) in sorghum, Sotuba, Bamako (Average 2011–2014).

| Treatments | No. Hills ha$^{-1}$ | Diam. Straw 30 DAS mm | Diam. Straw 50% Flowering mm | Stover kg ha$^{-1}$ | Grain kg ha$^{-1}$ | Gross Margin FCFA ha$^{-1}$ | VCR | FUE |
|---|---|---|---|---|---|---|---|---|
| Control | 18,160 a | 9.2 ab | 24 a | 10,200 a | 1174 a | 97,450 a | - | - |
| SP | 17,720 a | 7.6 a | 24 a | 9328 a | 1171 a | 97,130 a | - | - |
| SP + ST | 18,780 a | 11 b | 28.6 a | 12,580 a | 1264 a | 106,400 a | - | - |
| SP + ST + M1 | 18,630 a | 10 ab | 23 a | 11,420 a | 1380 a | 118,100 a | 15.2 a | 41.00 a |
| SP + ST + M1 + M2 | 19,220 a | 11 b | 22.4 a | 12,850 a | 1526 a | 132,600 a | 14 a | 34.60 a |
| SP + ST + M1 + UR | 19,100 a | 11.2 b | 23.8 a | 17,100 b | 2068 b | 186,900 b | 12.8 a | 29.60 a |
| SE + | 551.09 | 1.03 | 2.49 | 1837.83 | 185 | 16,710 | 9.96 | 26.4 |
| Probability | 0.40 | 0.13 | 0.58 | 0.21 | 0.3 | 0.007 | 0.7 | 0.74 |

SP = seed priming, ST = seed treatment, M1 = 5 kg NPK ha$^{-1}$ at sowing, M2 = 5 kg NPK ha$^{-1}$ at tillering, UR = 25 kg urea ha$^{-1}$. Numbers followed by different letters are statistically different according to Duncan multiple rank test. DAS = days after sowing, FCFA = West African CFA franc.

In Koporo-Pen in central Mali, stover and grain yield of pearl millet also increased with increasing level of intensification (Table 9). Here, the treatment that included seed priming, seed treatment and microdosing increased stover and grain yield compared to the control by 1480 kg ha$^{-1}$ (102.7%) and 424 kg ha$^{-1}$ (77.7%) respectively. This treatment also increased gross margin over the control by 122.6%. It was particularly microdosing that gave the yield-enhancing effect at this site. Contrary to Sotuba, there was no effect of top dressing of fertilizer on yield. The treatment that included seed priming, seed treatment and microdosing gave a VCR of 61.3 that is way above the critical limit of four.

**Table 9.** Effect of increasing level of intensification on stover yield, grain yield, gross margin, value–cost-ratio (VCR) and fertilizer use efficiency (FUE) in pearl millet, Koporo-Pen, Mopti region. (Average 2011–2014).

| Treatment | Stover kg ha$^{-1}$ | Grain kg ha$^{-1}$ | Gross Margin FCFA ha$^{-1}$ | VCR | FUE |
|---|---|---|---|---|---|
| Control | 1440 a | 546 a | 34,590 a | - | - |
| SP | 1673 a | 584 a | 38,440 ab | - | - |
| SP + ST | 1977 a | 672 a | 47,210 abc | - | - |
| SP + ST + M1 | 2920 b | 970 b | 76,990 d | 61.25 b | 211.5 b |
| SP + ST + M1 + M2 | 3010 b | 837 b | 63,720 bcd | 25.25 a | 72.5 a |
| SP + ST + M1 + UR | 2440 b | 903 b | 70,290 cd | 12 a | 29.5 a |
| SE ± | 213.1 | 89.1 | 8909.4 | 7.77 | 25.96 |
| Probability | 0.0001 | 0.014 | 0.014 | 0.0001 | 0.0001 |

SP = seed priming, ST = seed treatment, M1 = 5 kg NPK ha$^{-1}$ at sowing, M2 = 5 kg NPK ha$^{-1}$ at tillering, UR = 25 kg urea ha$^{-1}$. Numbers followed by different letters are statistically different according to Duncan multiple rank test. FCFA = West African CFA franc.

## 4. Discussion

This study showed that it was feasible to intensify sorghum and millet cultivation using low-input agricultural technologies in Mali. However, the way to intensify millet and sorghum cultivation was dependent on the climatic zone and on the crop grown.

### 4.1. Seed Priming and Seed Treatment with Fungicide/Insecticide

In southern areas of Mali, both Experiments 1 and 2 confirmed that there is no effect of seed priming on sorghum and pearl millet. However, Experiment 1 showed that seed priming increased grain yield compared to the control in the low rainfall of sites of Madina Kagoro, Koporo-Pen and Koro by 19.7%, 26.6% and 18.0% respectively. Experiment 2 also showed a tendency that seed priming increased pearl millet yield in Koporo-Pen. Previous research has shown that seed priming increased

millet yield with 40% in the Mopti region [22] while seed priming in Sudan increased sorghum grain yield by 67.4%. Seed priming is known to improve crop establishment and plant vigour, advance flowering and increase yield under dryland conditions [14]. Seed priming was also found to greatly increase the economic return to farmers. Gross margin increased as a result of seed priming compared to the control in Madina Kagoro, Koporo-Pen and Koro by 18,750 (37.5%), 21,010 (56.2%) and 10,970 FCFA ha$^{-1}$ (66.9%) respectively. This increase was a result of increasing both grain and stover yield. These results showed that seed priming is a very promising entry point for intensification in the Sahel. Even though seed priming did not increase yield and profitability in southern Mali, it can still be an option under conditions of delayed or erratic rainfall during the sowing period.

There was no statistically significant effect of treating the seeds with a combined fungicide/insecticide in Experiment 2. However, there was a tendency that seed treatment increased yield in both sites. For sorghum in Bamako, the yield increase of the treatment compared to the control was 93 kg ha$^{-1}$ (7.9%) while in Koporo-Pen this yield increase was 88 kg ha$^{-1}$ (15.0%). Previous research has shown that yields of sorghum and millet increased on average by 17% as a result of seed treatment [11]. This treatment may also act as an assurance in the case of severe attack of pests and diseases at the crop establishment phase. Use of a planter can make it possible for the farmer to sow without direct contact with treated seeds.

### 4.2. Microdosing

Experiment 1 and Experiment 2 clearly showed that higher rates of microdosing are justified in southern Mali where rainfall and yield is higher than in central areas with lower rainfall and yields.

In sorghum (Experiment 1), there was an increasing response to microdosing up to 20 kg ha$^{-1}$ in Sotuba while for pearl millet it was only the application of 20 kg NPK ha$^{-1}$ that significantly increased grain yield compared to the control.

The FUE efficiency of 20 kg NPK ha$^{-1}$ for sorghum and pearl millet was 31.6% and 9% respectively. A fertilizer use efficiency of 10 and above is considered efficient [16]. These results illustrate a lower response to microdosing in pearl millet than in sorghum. These effects of microdosing were also a result of the low levels of soil organic carbon and of available phosphorus in these soils (Table 2). Results from north Kordofan in Sudan also confirmed that sorghum responds more to microdosing than pearl millet [11].

In central Mali with less rainfall, even the lowest rates of microdosing increased the yield in sorghum and pearl millet. Sorghum responded very positively to microdosing in Madina Kagoro. In Koporo-Pen and Koro, the application of 4 kg NPK ha$^{-1}$ increased grain yield compared to the control by 39.7% and 23.7% respectively. However, at these sites there was no effect of applying rates beyond 6 kg NPK ha$^{-1}$. Experiment 2 also showed that 0.3 g NPK hill$^{-1}$ can greatly increase yield in Koporo-Pen. In Koro, the highest fertilizer rate gave a lower FUE than the lower microdosing rates. The optimal microdosing rate for the Mopti region seemed to be 4 kg NPK ha$^{-1}$ (0.4 g NPK hill$^{-1}$) as this rate gave the highest VCR ratio and a high gross margin.

Experiment 2 showed that the application of 25 kg urea ha$^{-1}$ as top dressing at tillering increased sorghum yield in Sotuba by 688 kg ha$^{-1}$ (49.9%). This effect is in line with previous studies by showing that the relative importance of N compared to P is enhanced with increasing rainfall [3]. In central Mali with less rainfall, there was no effect of top dressing with urea. Higher fertilizer rates are therefore needed in the higher rainfall areas as in Sotuba, Bamako because the yield level is higher. The yield in the control treatment varied between 500 and 700 kg ha$^{-1}$ in Koporo-Pen and Koro while it was beyond 1000 kg ha$^{-1}$ in Sotuba, Bamako.

Most of the research on microdosing has previously focused on application of microdosing rates between 2 and 6 g fertilizer hill$^{-1}$ and such rates have been found to increase the grain yield of sorghum and millet between 28.7% and 107.5% [7]. Rates below 0.6 g hill$^{-1}$ have been able to increase yield in the order of 31.3%–68.7% [7]. Recent research shows that rates between 2 and 6 g NPK hill$^{-1}$ may not give a satisfactory result for many farmers. In Niger it was shown that 36% of 276 demonstration

plots had a VCR of below one when a microdosing rate of 2 g DAP hill$^{-1}$ was used [12]. The minimum VCR is three to four in high risk production environments [16]. It has furthermore been shown that microdosing is generally not profitable in labour-scarce environments [23], because of the workload of applying microdosing in a separate operation.

It may appear difficult to explain why rates as low as 4 kg NPK ha$^{-1}$ can give a yield-enhancing effect, but a study on seed coating by ICRISAT showed that 0.1 kg P ha$^{-1}$ applied as seed coating was able to increase pearl millet dry matter biomass 20 days after planting by 280% and grain yield by 48% compared to a control without phosphorous application [21]. Furthermore, seed coating increased the amount of P in the plants at tillering by three times compared to the control treatments. Small seeded cereals like pearl millet need an early supply of nutrients as the plants will exhaust the seed reserves of nutrients 8–10 days after sowing and all floral parts are formed 30 days after sowing [24]. Even though an application of 0.1 kg P ha$^{-1}$ seems very low, this amount represents 18 times the amount of the phosphorus reserves in seeds at sowing. Microdosing has also been found to increase leaf chlorophyll concentration at the tillering stage and to promote early lateral root development [25]. In the seed coating experiment, the rate of 0.1 kg P ha$^{-1}$ was used [24], but the rate of 4 kg NPK ha$^{-1}$ in Mali corresponded to 0.6 kg P ha$^{-1}$. Therefore, there are good reasons to believe that such low microdosing rates will facilitate crop establishment and better tillering.

### 4.3. Intensification of Millet and Sorghum Production

Given the rapidly growing demand for food in the drylands of West Africa, there is a need to increase food production [1]. This study showed that grain yield could be increased up to 124% by introducing the yield-enhancing technologies of seed priming, seed treatment, microdosing and top dressing with urea. These technologies are also highly profitable as evidenced by their favourable gross margins (up to 122% increase) and a high value–cost-ratio. These rates and technologies will not realize the full yield potential of the crops, but instead, they represent entry points for agricultural intensification.

The sustainability of microdosing has recently been questioned as it can lead to mining of plant nutrients [26]. However, the sustainability of the microdosing method cannot be assessed based on the nutrient balance alone; it needs a holistic assessment considering socio-economic and institutional factors. A recent study from Mali showed that the use of microdosing creates a surplus cereal production [27]. Part of the income generated from the sale of the surplus production can be used to purchase fertilizer. It has furthermore been reported from Mali that farmers who practise microdosing purchase more animals and invest in more farm equipment [28]. Farmers who practice seed priming and microdosing will therefore over time be able to better address the problem related to negative nutrient balance, as more organic and mineral fertilizer will be available to the farmers. We therefore think it is misleading to judge the sustainability of a practice based only on the nutrient balance at the plot level, because the overall effect of a technology on the nutrient balance depends on the ability of the technology to create an economic surplus and on how farmers use this economic surplus.

There are different views on how fertilizer microdosing should be practised despite continued research on microdosing since the 1990s. ICRISAT has recommended applying 2 g DAP hill$^{-1}$ or 6 g NPK hill$^{-1}$. Our research showed that there are good reasons to modify this general recommendation taking into consideration rainfall, crop grown, crop density, degree of mechanization, access to organic fertilizer and farmers' preferences. A decision tree for crop and fertilizer management that includes rainfall, crop choice, use of priming and microdosing rates as decision variables was developed based on the results of the experiments included in this study (Figure 1). Technologies that should be promoted independent of these factors include appropriate varieties, seed quality, treatment of seed with a combined fungicide/insecticide, integrated pest and weed management and use of organic inputs according to farmers' access. This integrated package will also ensure the long-term sustainability of the system.

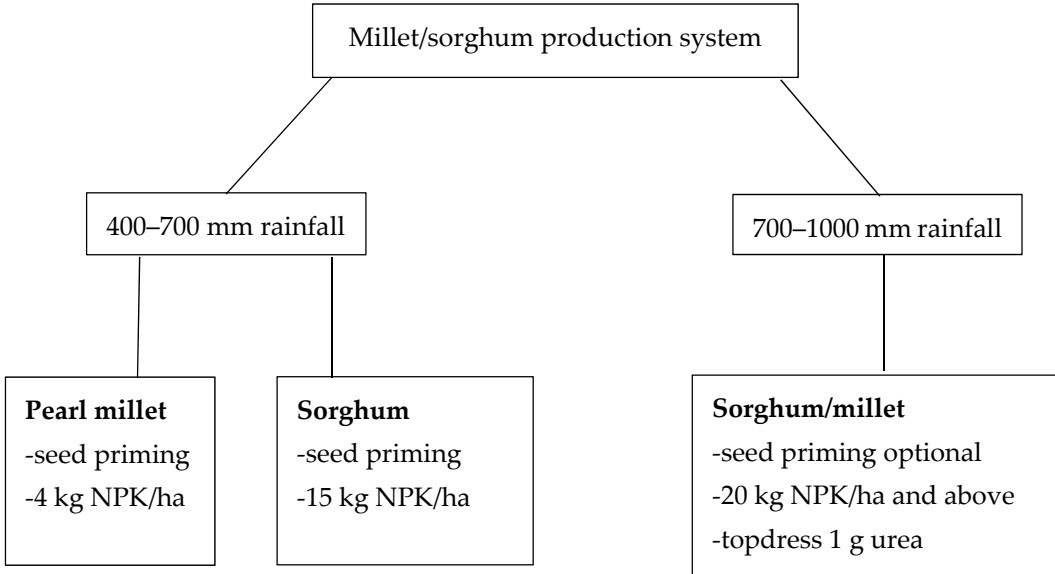

**Figure 1.** Decision tree for crop and fertilizer management in sorghum and pearl millet.

Rainfall is the first management differentiating factor in the decision tree, as we have shown that rates as low as 4 kg NPK ha$^{-1}$ can be used in areas receiving 400–700 mm rainfall, while rates should be at least 20 kg NPK ha$^{-1}$ in areas with rainfall beyond 700 mm. Our results also demonstrated that choice of crops matters. Sorghum appears to respond more to microdosing than pearl millet, as the highest yield increase in sorghum in Experiment 1 was 638 kg ha$^{-1}$ while in pearl millet the highest increase was only 187 kg ha$^{-1}$. This means that, for cash constrained farmers who cultivate both pearl millet and sorghum but cannot afford to fertilize both crops, the emphasis should be on microdosing of sorghum and less on pearl millet.

Farmers may also consider the use of a donkey-drawn planter/weeder as part of the intensification process. If farmers choose to apply a rate of about 4 kg NPK ha$^{-1}$ (0.4 g hill$^{-1}$), it is possible to mix seed and fertilizer in an approximately 1:1 ratio in the hopper of the planter. However, if farmers choose to apply 2 g fertilizer hill$^{-1}$ and beyond, they cannot mix seed and fertilizer because that would result in burning of the seed. Most planters used in the Sahel for sorghum and millet do not have a separate hopper for fertilizer; therefore, in practice, it is impossible to use fertilizer rates of 2 g fertilizer hill$^{-1}$ and beyond in mechanized sowing. Use of a combined donkey-drawn planter/weeder can make these technologies more appealing because labour demand is consequently reduced by 89% [13]. The planter/weeder can ensure optimal sowing time, appropriate sowing depth and plant density, uniform seed and fertilizer rate [13]. The farmers can, in addition, avoid direct contact with seeds treated with fungicides/insecticide. Mechanization alone has been shown to increase yields by 14% in sorghum in Mali [13].

Agricultural intensification in Mali and other Sahelian countries is difficult because insecurity prevails in the region. The low-input technologies proposed here will not unnecessarily expose the farmers to additional risk as the cash outlay is very low and with a low risk of crop failure. In none of the four years did the improved technologies give a lower yield than the control (farmers practice).

## 5. Conclusions

This study showed that crop and soil fertilizer management in the drylands of West Africa should be differentiated according to rainfall and the crop grown. In central Mali, with rainfall ranging from 400 to 700 mm, we showed that seed priming in combination with 4 kg NPK ha$^{-1}$ gives the best result in terms of yield and VCR. In areas further south with 700–1000 mm rainfall, farmers may use 5 kg NPK ha$^{-1}$ applied as microdosing at sowing combined with top dressing of 25 kg urea ha$^{-1}$ at tillering. Higher fertilizer rates are needed in the south compared to areas in central Mali owing to higher yield

levels in the areas with more rainfall. Sorghum was more responsive to microdosing than pearl millet. Seed priming was able to increase yield in central Mali, but not in southern Mali. The gross margin and value-cost-ratio analysis showed that these technologies can greatly improve profitability of sorghum and pearl millet production in the drylands of Mali. The technologies identified in this study can therefore be considered as entry points for sustainable agricultural intensification in the drylands of West Africa.

**Author Contributions:** A.C. and K.W. participated in the conceptualization, methodology, formal analysis, investigation and writing—reviewing and editing. J.B.A. participated in conceptualization, methodology, formal analysis and writing—original draft preparation.

**Funding:** This research was funded by the Norwegian Ministry of Foreign Affairs through the project "Adapting crop and livestock to climate change in Mali".

**Acknowledgments:** We would like to thank staff at Sotuba and Koporo-Pen research stations for facilitating this work.

**Conflicts of Interest:** The authors declare no conflict of interest.

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
