# Peer review of "Sustainable Intensification of Sorghum and Pearl Millet Production by Seed Priming, Seed Treatment and Fertilizer Microdosing under Different Rainfall Regimes in Mali"

_agronomy, doi:10.3390/agronomy9100664_

Round 1

Reviewer 1 Report

I thank the authors very much for carefully addressing all of my comments! In my opinion, the authors have successfully revised the manuscript. Best wishes for the future!

Reviewer 2 Report

The manuscript have been thoroughly revised .

This manuscript is a resubmission of an earlier submission. The following is a list of the peer review reports and author responses from that submission.

Round 1

Reviewer 1 Report

Dear Authors,

please find my major and minor comments (117 in total) within the annotated PDF attached to this report.

Furthermore, I would strongly recommend providing some figures (e.g. bar charts) for presenting the most important results in the revised manuscript. In its present form, it is quite exhausting to read the results (too many tables).

Kind regards.

Author Response

Rebuttals reviewer 1

Line 1-2. I think that a descriptive title is a good idea for this paper. However, it must be reformulated, because in this form it does not fit to the content of the paper. For example, this study focuses on the optimization of the sowing procedure. This study is not on the overall approaches to intensify sorghum / millet cultivation. Thus, both this sentence and the title must be reformulated using terms like 'sowing procedure', 'seed priming management', 'crop and soil fertilizer management' and 'seed-fertilization' (not all of them, I only gave some examples that may help improving the title).

Reply: We suggest to keep the term “sustainable intensification” as we have described how the technologies tested contribute to sustainable intensification. However, we have formulated a new title.

New title suggested: Sustainable intensification of sorghum and pearl millet production by seed priming, seed treatment and fertilizer microdosing under different rainfall regimes in Mali

Line 6: Then, affiliation 3 must be renumbered (2), of course.

Reply: Corrected

Line 12. One sentence about the broad background of the research objective must be added here (why is it important to grow sorghum and millet, why does it has to be done more sustainably? etc.).

Reply: It is difficult to cover these aspects to any depth in the abstract as the space is very limited. However, a new sentence is added in order to better explain the context.

Line 14: Hill.don't know what this measn? Does it refer to the plots? Must be explained when first mentioned in both the abstract and the main body of the manuscript.

Reply: Seeds are placed in clusters which are called hills. This is the term used in the scientific literature to describe how seeds are millet and sorghum seeds are sown in West Africa. The term is used in major scientific journals like Field Crop Research, Experimental Agriculture and Journal of Nutrient Cycling. The papers that we have given reference number 4, 7 and 22 also use this term. We explained the term as follows in the introduction:

Line 16: Does this refer to the sorting of the input listed within the brackets in the middle of this sentence? Then it is not required and should be omitted.

Reply: The sentence is reformulated as follows: ..experiment 2 assessed the cumulative effects of the inputs seed priming, seed treatment with pesticide, microdosing and urea top dressing.

Line 18: Empty space

Reply : Empty spaced is removed.

Line 20: Also referring to microdosing? Must be formulated more concisely.

Reply: The sentence is reformulated as follows:

the fertilizer microdosing rate of 0.8 g NPK hill-1

These are six important key factors that must be referenced!

Reply: New references are given (see references)

Line 37. Price of input: This factor must be presented in an own sentence, because the price of fertilizer is very important in context of the topic of this study.

Reply: Farmers face many constraints, and we consider that price of input if one of these factors as explained in the text. We therefore do not think it is necessary to add any additional sentence about this. The price of input is taken into consideration in the economic analysis.

Line 39: Prior to this sentence, a comprehensive list of examples for low-input agriculture must be provided, including standard low-input measures such as low tillage, low weeding intensity, low irrigation intensity etc. Each of them must be referenced using original literature as well.

Reply: We give examples of such low input technologies in the text below with original references

Line 41: ICRISAT. I don't know what this is. Please explain this abbrevation when first mentioned!

Reply: ICRISAT is the International Crop Research Institute for the Semi-Arid Tropics.

The full name of the institute is presented in the text.

Line 43: NPK. Again, abbrevations must be explained when first mentioned.

Reply: NPK stands for N (nitrogen) P (phosphorous) K (potassium). I think readers of the journal will be familiar with the term NPK.

Line 44: Here, the main actors of this study, the crops pearl millet and sorghum, are first mentioned. They must be introduced much more comprehensively in an own paragraph (using original literature, of course). For example, where do they come from, what are their botanical names, how much do they contribute to human nutrition in Niger, how are they cultivated (demands for soil quality, precipitation/irrigation, seed costs, sowing depth etc.), what are problems in cultivating them, and why is fertilization so important for their growth?

Reply: New text is inserted to better address these issues.

Line 51; delete. Write 'by' instead.

Reply: Done.

Line 51: A study in Niger concluded. In a study conducted in Niger, it was conclude that (...)

Note, that studies cannot actively conclude but include conclusions

Reply: Reformulated as follows:

It was concluded based on a study in Niger…

Line 52-54: I don't understand this sentence, please reformulate and make sentences shorter!

Reply: The sentence is reformulated.

Line 61. Low hanging fruit. Reformulate. Consider using the term 'agricultural low-input practice'. PLease also see a comment above.

Reply: We have changed as suggested by the reviewer.

Line 64. Reference missing.

Reply : Reference included. Harris 2006.

Line 65-67. Reformulate - consider two sentences (keep sentences short!).

Reply : The sentences are cut in two as suggested.

Line 61: hill. Use SI-units. See above.

Reply: There is no SI unit which corresponds to hill or cluster of seeds. We have described the meaning of a hill. Hill is a common term used in the scientific literature on cereal production in the drylands of West Africa.

Line 103: use the term 'geometry' instead of 'distance' here.

Reply: The term distance is replaced with geometry.

Line 108: Do you mean 'sampling area'? Then use this term instead of 'harvested area'.

Reply: We think it is more appropriate to use harvest area instead of sample area because harvest area better describe what was done. We did not take only samples.

Line 111: NPK Provide product name, company name

Reply. New sentence formulated:

The fertilizer used was the compound and granulated NPK 15-15-15 (produced by STPC)

Line 119: 'increasing fertilization' not 'intensification'

Reply: I suggest to keep the word “intensification” here because this study in not only about increasing levels of fertilization, but it is also about seed priming and treatment of seeds with pesticides.

Line 13. Please provide comprehensive description of the statistical model used for statistical analyses of both experiment 1 and 2!

Reply: This is now included in text.

Line 144: Link to Table 2?

Reply: This is done.

Line 155-157. Please remove this section to the results section.

Reply: This aspect is now included at the end of the discussion

Table 3: DAS. Explain abbrevation when first mentioned!

Reply: DAS means “days after sowing”. The term explained below the tables.

Table 3 and other tables. Use either 'XX XXX' or 'XXXXX' or XX,XXX.X' but not a mix-up. see comment above.

Reply: I assume that the final formatting of the tables will be done by the journal.

Table 3: Fcfa. What does this mean?

Reply: This means “West African CFA franc”. This is explained below the table.

Line 178. where? Within treatments? within parameter? Across treatments? Please define more precisely.

Reply: The new text is reformulated as follows. Numbers followed by different letters are statistically different treatments according to Duncan multiple rank test.

Line 279. increase fertilization level

Reply: I suggest to keep the term “intensification” because this is not only about fertilizer levels, but it is also about seed priming and treatment of seeds with pesticides.

Line 279 using agricultural low-input practices. Please see comment above (Biala et al.).

Reply: New sentence formulated as follows: using low-input agricultural practices. Please see comment above (Biala et al.).

Line 279: replace cultivation by production

Reply: done

Line 280: climatical zone?

Reply: climatic zone included

Line 308-312. I think, this is too much of repeating the results. Please reconsider what is really important to be mentioned within the discussion section.

Reply: Sentence shortened and reformulated. See text

Line 320 -327. I think, this is too much of repeating the results. Please reconsider what is really important to be mentioned within the discussion section.

Reply: Paragraph shortened. See changes in the text.

Line 332 Schlecht et al. 2006 delete?

Reply: We think this reference should be there.

Line 335: I think, this is too much of repeating the results. Please reconsider what is really important to be mentioned within the discussion section.

Reply: This sentence is important in order to explain why there is better response to fertilizer in southern Mali.

Line 349. Furthermore, the seed coating (...)

Reply: The sentence is changed as proposed.

Line 357. Reformulate, e.g.: 'This indicates, that low microdosing rates (...)

Reply: Sentence reformulated

Line 368-379. Wrong format

Reply : Reformulated

Line 381: Wrong format

And the figure is of poor quality, please improve the optical appearance of the figure.

Reply: New figure inserted. Please inform if a specific format is required to produce the figure.

Line 383: 'fertilizer microdosing'?

Reply: change included.

Line 416: it was shown

Reply: changed as suggested

Line 436: The reference section is not well-formatted and needs re-formatting.

Furthermore, the references must be more comprehensive (at least 40 references, in my opinion). The whole manuscript must be revised under this aspect.

Reply: Since this is not a review article, we do not think as many as 40 references are need. We have used the number of references we think are needed to document the statements. However, we have added 3 more references in connection with some of the statements in the paper. The references have been checked and minor errors corrected.

Reviewer 2 Report

Understanding the optimal fertilization rates under varius climatic zones is of critical importance in crop production. Therefore this work is of paramount importance for regions worldwide facing problems relating to food security. The topic of this work is closely falls within the aims and scope of the journal. The data provided are sufficient and the statistical analysis of the results is very well presented. Tables clearly present the data. The discussion of results focus on the main points while justification of the findings are well supported by references. A few modifications in Section “Materials and Methods” are suggested. Please see the attached document. 

Author Response

Rebuttals for reviewer 2

Line 86. Please insert horizontal lines in Table 1.

Reply: We are not sure the editor will accept horizontal line within a table. However, we have reduced the font to make the table easier to read.

Line 97. Better add this information in Table 1.

Reply: We find it difficult to insert more information in Table 1.

Line 25. Please use a table and explain the initials SP ST M

Reply: Abbreviations are explained in the text.

Line 250: Please leave a space

Reply; An additional line is added give extra space.

Line 253: It would be better if you can present the treatments with cod names

Reply: Treatments are presented with code names.